# Short-Term Postoperative Outcomes between 4% Icodextrin Solution and Hyaluronic Acid- Carboxymethyl Cellulose Membrane during Emergency Cesarean Section

**DOI:** 10.3390/jcm8081249

**Published:** 2019-08-18

**Authors:** Kuan-Sheng Lee, Jian-Pei Huang

**Affiliations:** 1Department of Obstetrics & Gynecology, MacKay Memorial Hospital, Taipei 104, Taiwan; 2MacKay Junior College of Medicine, Nursing and Management, New Taipei City 251, Taiwan; 3Department of Medicine, MacKay Medical College, New Taipei City 252, Taiwan

**Keywords:** emergency cesarean section, seprafilm, adept, contaminated wound, dirty/infected wound, metritis

## Abstract

Emergency cesarean sections are associated with more postoperative complications than with elective cesarean sections. Seprafilm and Adept are commonly used adhesion reduction devices and have been applied in abdominal or pelvic surgery for a long time. This study focuses on comparing the short-term postoperative outcomes of emergency cesarean sections between two groups. We performed a retrospective study that included all patients who received emergency caesarean sections from the same surgeon at MacKay Memorial Hospital between August 2014 and November 2017, We analyzed the overall cases and conducted a subgroup analysis of cases with contaminated or dirty/infected wounds in regard to the rates of surgical-site infection (SSI), bandemia, delayed flatus passage, and length of hospital stay. The two groups were similar with respect to the rates of SSI, bandemia, and length of hospital stay. However, Seprafilm was associated with higher risk of delayed flatus passage over 48 h (OR: 2.67, 95% CI = 2.16–7.64, *p* = 0.001). It also needs less time for recovery of the digestive system and less medical management postoperatively. In cases of contaminated or dirty/infected wounds, Adept user also had significantly lower rates (10.3% vs. 32%, *p*
*=* 0.048, OR: 4.12, CI = 1.09–15.61) of postcesarean metritis.

## 1. Introduction

Emergency cesarean sections have higher rates of postoperative surgical-site infection (SSI), prolonged hospitalization, fever, and urinary tract infection than elective cesarean sections [1]. Perioperative complications (e.g., ileus, metritis, and wound complications) increase postcesarean maternal morbidity, increase health costs, and cause longer hospital stays [2,3,4]. Icodextrin 4% solution (Adept) and hyaluronic acid-carboxymethylcellulose membrane (Seprafilm) are common adhesion reduction devices that are used worldwide. Their efficacy has been proven in multiple studies, but most research has been conducted on elective surgeries [5,6,7,8,9]. In our experience, some Sepraflim users have experienced delayed flatus passage and more febrile episodes that may require therapeutic antibiotics during the postcesarean period.

In our opinion, Seprafilm may be a barrier that prevents the omentum from playing its protective and immunological role to absorb the microabscess and decrease the local infection at an emergency cesarean surgical site. In contrast, Adept is used for irrigation, which may reduce debris, bacterial colonies, and blood clots before closure of the wound. Therefore, it may provide a solution to reduce complications after emergency cesarean sections. Thus, we retrospectively compared the postoperative outcomes, especially postcesarean SSI, between Seprafilm and Adept during emergency cesarean sections and try to prove our hypothesis.

## 2. Methods

Data were collected by staff for all emergency caesarean sections performed by the same surgeon at MacKay Memorial Hospital between August 2014 and November 2017, Reduction of adhesions in pelvic and abdominal surgery including cesarean section is the indication of Adept and Seprafilm in Taiwan. Therefore, we provide Adept and Seprafilm, as a choice after discussion, to our patients before undergoing operation.

Hospital records were used to obtain data on demographic and operation factors, including age, BMI, systematic disease, gestational disease, and wound condition around postoperative day 30 at the surgeon’s clinic. The data were excluded in the following cases: (a) American Society of Anesthesiologists (ASA) score >3; (b) inability to give/understand informed consent about operation and medical appliances or against-advise discharge; (c) severe immunosuppression; (d) superficial incisional wound evaluation excluding patients who were lost to follow-up visits arranged about 30 days after operation; and (e) patients needing reoperation. Finally, 287 patients were collected: 139 patients in the Adept group and 148 patients in the Seprafilm group.

Preoperative skin preparation, antibiotic prophylaxis or therapy, and surgical procedures were performed according to local standards. After closure of the uterus, debris, bloody fluids, and blood clots were removed as completely as possible. In the Adept group, the pelvic cavity was irrigated with about 500 mL of Adept, and the remaining 1000 mL was left in the abdomen. In the Seprafilm group, Seprafilm was placed over the anterior surface of the uterus, including the incisional site. Then, skin closure was performed using subcuticular sutures, and sterile dressing was applied without any further wound-related procedures. The Adept and Seprafilm were purchased, stored, and distributed according to the respective standards operating procedures of MacKay Memorial Hospital.

Due to higher infection rates, a subgroup analysis was conducted on cases with contaminated (e.g., membrane rupture for more than 18 h) and dirty/infected wounds (clinical or subclinical chorioamnionitis) which need preoperative or postoperative therapeutic antibiotic treatments [10].

Contamination of the surgical wound was classified using an adaptation of a standard definition [11] to include membrane rupture before operation. Hence, when membranes had ruptured for less than 18 h before the caesarean section, the wound was classified as clean-contaminated. If the membrane ruptured for more than 18 h, the wound was classified as contaminated. Clinical or subclinical chorioamnionitis was classified as dirty/infected.

Surgical site infections are generally defined as infections that occur after surgery in the part of the body where the surgery took place. These infections are classified as incision or organ/space infections (e.g., metritis, parametritis, peritonitis). Metritis (also called endometritis, endomyometritis, or endomyoparametritis) following cesarean delivery is historically referred to as puerperal fever with additional signs of uterine tenderness or parametrial tenderness, leukocytosis ranging from 15,000 to 30,000 cells/μL, and foul-smelling lochia [12,13]. Incisional wound infections included those involving only skin and subcutaneous tissue (superficial incision) and those involving the deeper soft tissues of the incision, such as muscle or fascia [14].

Factors recorded and analyzed as confounding factors were cases that required medical management (e.g., sip water, menthol packing, primperan, or dimethicone) to promote intestinal peristalsis or improve abdominal fullness, as well as preoperative or postoperative antibiotics or medication with effects on wound healing (e.g., immunosuppressive agents). The primary outcome measure was the incidence of SSI within postoperative day 30. Organ SSI (e.g., metritis) and incisional SSI were assessed clinically by the surgeon or an assistant during the inpatient stay, in the emergency room, or at the surgeon’s clinic around postoperative day 30 based on clinical criteria [15]. The secondary outcome was the rates of prolonged flatus passage (>48 h postcesarean), length of hospital stays, and the incidence of leukocytosis and bandemia. We compared leukocytosis and bandemia between the two groups on postcesarean day 1 based on routine laboratory data.

## 3. Statistical Analysis

Statistical analysis was performed with R software, version 3.3.1 (R Project for Statistical Computing, Vienna, Austria). Demographic and clinical data were compared between groups, a Student’s *t*-test, and the results for continuous variables are given as the mean ± the standard deviation. The odds ratios and corresponding 95% confidence intervals (CIs) were calculated to assess the effect of the different adhesion prevention barrier outcomes and the postcesarean outcomes using a logistic regression analysis with adjustment for relevant significant variables. Statistical significance was defined at the 95% level (*p* < 0.05).

## 4. Results

### 4.1. Demographic Characteristics

The characteristics of the 139 Adept and 148 Seprafilm subjects are presented in Table 1. There are no significant differences between groups in all the demographic characteristics, including history of abdominal surgery, the proportion of rupture of the membrane more than 18 h, prolonged labor before operation, and other results except for operation time and blood loss. The average operative time was 4 min longer in the Adept group (Seprafilm: 84 min, Adept: 88 min, *p =* 0.026). The blood loss was greater in the Seprafilm group (Seprafilm = 344 c.c., Adept = 282 c.c., *p =* 0.011). There were no significant differences between the two groups with respect to indications of emergency cesarean section (Table 2).

### 4.2. Analytical Outcomes between Seprafilm and Adept Groups

Overall, there was no significant difference between the two groups with respect to puerperal fever (Seprafilm vs. Adept: 7.4% vs. 5.8%, *p =* 0.568) and postcesarean metritis-related febrile episodes (Seprafilm vs. Adept 5.4% vs. 2.9%, *p =* 0.285) (Table 3). However, in the groups with contaminated and dirty/infected wounds, the Adept group had significantly lower rates of postcesarean metritis than the Seprafilm group (10.3% vs. 32%, *p =* 0.048, OR: 4.12, CI = 1.09–15.61) (Table 4). The difference that was more obvious after adjustment for confounders (e.g., age, DM, GDM, SLE, antibiotics, blood loss) [4,13,16] (*p =* 0.015, OR: 12.92, CI = 1.64–101.53) (Table 5).

There was no significant difference in the incisional wounds’ infection rates (6.1% vs. 5.8%, *p =* 0.789) and length of hospital stay (4.14 vs. 4.13 days, *p =* 0.852) between the Seprafilm and Adept groups (Table 3), even in the group with contaminated and dirty/infected wounds (incisional wound infectious rates: 12.0% vs. 10.3%, *p =* 1; length of hospital stay: 4.33 vs. 4.28 days, *p =* 0.827) (Table 4). At MacKay Memorial Hospital, we routinely check hemograms on postcesarean day 1. The published upper limit of the normal band at MacKay memorial hospital is 6% bands. We used a 10% upper limit of normal to ensure clinical significance and to match accepted sepsis definitions [17,18]. There was no significant difference in leukocytosis and bandemia between the two groups (2% vs. 0%, *p =* 0.248) or in the groups with contaminated and dirty/infected wounds (4% vs. 0%, *p =* 0.391) (Table 4). 

The Seprafilm group had longer flatus passage (>48 h postcesarean) compared with the Adept group (39.2% vs. 20.1%, *p* < 0.001). Seprafilm was associated with a higher risk of delayed flatus passage over 48 h (OR: 2.67, 95% CI = 2.16–7.64, *p* = 0.001) (Table 3). The hazard ratio became more obvious after adjustment for medical management for relief of postcesarean abdominal fullness or promoting intestinal peristalsis (OR: 3.02, 95% CI = 1.70–5.36, *p* = 0.001) (Table 6). At MacKay Memorial Hospital, we routinely use rectal dulcolax suppository if flatus passage is delayed for over 48 h. There was a higher percent of cases needing rectal dulcolax suppository in the Seprafilm group than the Adept group (28.4% vs. 17.3%, *p* = 0.025). Despite the more frequent use of dulcolax in the Seprafilm group, there was still a significantly longer time required for digestive system recovery than in the Adept group (time of delayed flatus passage: 4.90 h, 95% CI = 2.16–7.64) (Figure 1).

## 5. Discussion

The most common sources of persistent fever after cesarean section are surgical site infections, urinary tract infections, and mastitis. The overall postpartum infection rate is 7.4% following cesarean section [13,19], which is similar to our result (Seprafilm vs. Adept: 7.4% vs. 5.8%, *p =* 0.568). Reported rates of postpartum endometritis vary between studies and depending on whether the data included postpartum surveillance. More recent studies have quoted rates of 1.5–5% for planned or emergency caesarean sections [20,21]. The metritis rates of both groups in our study (Seprafilm vs. Adept: 5.4% vs. 2.9%) are consistent with previous reports. Therefore, our results reliably demonstrated that the use of Adept for irrigation and postcesarean instillation in the group with contaminated and dirty/infected wounds was associated with lower rates of organ SSI.

The risk of SSI depends on three major factors. Firstly, the American Society of Anesthesiologists’ score reflects the patient’s state of health before surgery. Secondly, wound classification reflects the degree of contamination of the wound. Thirdly, the duration of the operation reflects the technical aspects of surgery. The SSI rate increases with increasing risk index score [22]. In our study, most patients’ ASA scores were 2, and only a small fraction was 3 due to poor control of gestational diabetes or type 2 diabetes, uncontrolled hypertension, or preeclampsia with severe features. All operations were performed by the same surgeon, and the average duration of the operation was longer in the Adept group by only about 4 min mostly due to different application technique of these two devices. Though the blood loss during operation was greater in cases using Seprafilm than Adept. However, all the blood was evacuated out of the abdominal cavity as complete as possible in all patients. There was only very small amount of residual blood in both groups, so we believe that it will not cause significant influence on the post-operative outcome. Thus, the major factor with an impact on SSI in the Seprafilm groups was the condition of the wound.

The uterine cavity is usually sterile before rupture of the amniotic sac. As a consequence of labor and associated manipulations, the amniotic fluid and uterus become contaminated with anaerobic and aerobic bacteria. Then, bacteria gain access to the pelvis via the surgical incision, resulting in uterine infection and parametrial cellulitis [12,13]. A previous study reported that the amniotic fluid obtained at cesarean delivery from women in labor with membranes ruptured for more than 6 h had bacterial growth, and an average of 2.5 organisms was identified from each specimen [12]. Some studies showed that increased infection rates and prophylactic antibiotics are associated with significantly lower rates of chorioamnionitis or endometritis in women with prolonged membrane rupture [23,24]. Thus, it is reasonable to define prolonged membrane rupture using contaminated wounds.

Chorioamnionitis complicates as many as 40–70% of preterm births with premature membrane rupture or spontaneous labor [25] and 1–13% of term births [26,27,28]. Twelve percent of primary cesarean births at term involve clinical chorioamnionitis, with the most common indication for cesarean in these cases being failure to progress, usually after membrane rupture [29]. Select factors independently associated with chorioamnionitis and their strength of association were summarized, and those with prolonged membrane rupture over 18 h have a higher risk for chorioamnionitis. Although there are no parameters associated with the length of membrane rupture before delivery to determine the wound class, we chose prolonged membrane rupture for over 18 h as the cut-off point for contaminated and dirty/infected wounds according to a previous summary [30].

Our results suggested that there is a correlation between the incidence of organ SSI and the different adhesion reduction devices in the subgroup analysis. The rate of fever caused by metritis was lower in the Adept group. Prompt initiation of antibiotic therapy is essential to prevent maternal complications in the setting of clinical chorioamnionitis [29,30,31,32,33]. Our results of length of hospital stay between groups had no clear difference. However, the use of therapeutic antibiotics for chorioamnionitis or other infections that affect postcesarean febrile rate were analyzed and adjusted for age, SLE, DM, GDM, and antibiotics, which showed a bigger difference.

We hypothesized that using Adept not only for adhesion reduction but also for irrigation could reduce postoperative organ surgical site infection. Debris, bacterial colonies, and blood clots are removed more from the wound when using Adept/suction and proper therapeutic antibiotics to help immune defenses and control infection. A previous animal study proved that this method works [34]. Currently, the 2016 WHO global guidelines for the prevention of SSI as well as the updated 2017 Centers for Disease Control and Prevention (CDC) guidelines [35] conclude that there is insufficient evidence to recommend routine intraoperative irrigation with saline. However, the level of underlying evidence is low, and trials that analyzed these guidelines do not solely focus on visceral surgery but include all types of surgery (e.g., orthopedic or neurosurgery), which differ substantially in SSI rates and causative microorganisms.

A large-scale meta-analysis of 41 RCTs on intraoperative irrigation with any solution (e.g., saline, PVP-I, or antibiotic solutions) showed a SSI risk reduction of 46% in the treatment group for abdominal surgery exclusively [36]. One standardized RCT comparing intraoperative irrigation with saline versus no irrigation after open appendectomies was published in 2000 and found a reduction of SSI from 25% to 8.7% in the saline irrigation group [37]. Studies concluding that intraoperative irrigation reduces SSI risk commonly used wound classifications of “contaminated” and “dirty/infected” (e.g., appendectomy studies). Thus, it seems that operation with class III and class IV wounds would have more benefits from intraoperative irrigation than class II wounds.

Two RCTs compared postoperative infections between Adept and lactated Ringer’s solution groups and mentioned no clear difference in the risk of SSI [38]. Thus, it is feasible to use Adept for irrigation and instillation. The wounds in major cases in our study were class II, and only 64 women (22.3%) were class III and class IV. Thus, there was no clear difference in postcesarean febrile episodes overall, but there was significant difference in the subgroup analysis on those with wound classifications of class III and IV, which is consistent with the RCT results.

There are several intraoperative preventive measures to reduce SSIs. Such techniques include removing devitalized tissues, appropriate antimicrobial prophylaxis, and antiseptic wound lavage [39,40,41,42,43]. Intraoperative wound irrigation of the subcutaneous and deep soft tissue before skin closure with saline or antiseptic solutions hypothetically are easy and economical options to reduce SSI rates [44].

Leakage of Adept from the abdominal cavity may moisten incisional wounds and increase infection rates. However, our results showed there was no difference in incisional wound infection rates between groups. This is reasonable because we suture the peritoneum tightly and make sure there is no leakage of Adept from the peritoneum cavity before closure of the skin incision.

The presence of bandemia was associated with an increased OR of having significant infection [18]. Our result showed three patients (2%) with elevated bandemia (≥10%) in the Seprafilm group and none in the Adept group. However, there was no statistical difference between the two groups. The small number of cases may have led to this result.

Our results also showed that the Seprafilm group needed more time for recovery of the digestive system after operation compared with the Adept group. One prospective RCT reported that small bowel obstruction and ileus are common adverse effects of Seprafilm. In this study, there was no statistically significant difference between the incidence of small bowel obstruction or ileus, as also reported in some studies [45,46]. However, our study showed that Seprafilm has a higher rate of prolonged flatus passage (>48 h) than Adept. Thus, we think that Adept may not only play a role as an adhesion reduction device but also may help quicken recovery of the digestive system after operation.

Our study has several limitations. First, it is a retrospective, chart-review, observational study for the comparison of postcesarean conditions between cases using Seprafilm and Adept. Thus, the results cannot be generalized to all abdominal surgeries. Lastly, it does not have as much power as an RCT.

## 6. Conclusions

Adept users have a lower incidence of postcesarean metritis in emergency cesarean sections with contaminated and dirty/infected wounds. It also needs a shorter recovery time of the digestive system and less medical management postoperatively. Thus, we believe that it is beneficial to use Adept rather than Seprafilm in emergency cesarean sections.

## Figures and Tables

**Figure 1 jcm-08-01249-f001:**
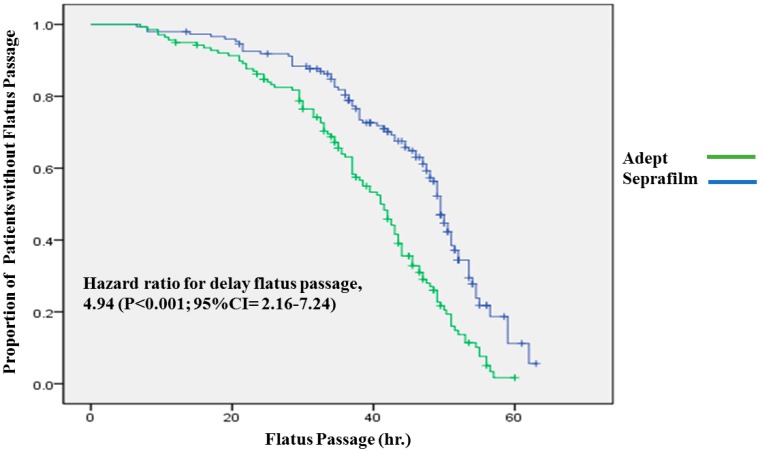
The time of flatus passage (hour). Kaplan–Meier plot of the time of flatus passage, measured from the end of operation to flatus passage. A Cox proportional-hazards model was used to determine the hazard ratio and 95% confidence interval. The Seprafilm group needed longer time (average 4.90 h) for digestive recovery than the Adept group.

**Table 1 jcm-08-01249-t001:** Baseline characteristics of the patients.

Characteristic	Seprafilm (*n* = 148)	Adept (*n* = 139)	*p*
Age	33.93 ± 4.03	34.47 ± 3.93	0.244
BMI	26.91 ± 3.68	27.21 ± 4.26	0.651
Gestation (weeks)	37.01 ± 3.21	36.87 ± 3.50	0.726
Production			0.929
*p* = 0	85 (58.1%)	84 (64.4%)	
*p* ≥ 1	62 (41.9%)	55 (39.6%)	
OP time (min.) ^#^	83.61 ± 17.31	87.86 ± 14.71	0.026 *
Blood loss (c.c.)	344.26 ± 200.21	282.01 ± 213.15	0.011 *
Types of Anesthesia			0.393
Spinal Anesthesia	137 (92.6%)	124 (89.9%)	
General Anesthesia	11 (7.4%)	12 (8.7%)	
Epidural Anesthesia	0 (0.0%)	2 (1.4%)	
Previous abdominal adhesion history	11 (7.4%)	6 (4.3%)	0.264
MR before C/S > 18 h	21 (14.2%)	31 (22.3%)	0.075
In labor before C/S	123	120	0.449
Use PCA	143 (96.6%)	129 (92.8%)	0.147

* *p* < 0.05. ^#^ Including time of preparation.

**Table 2 jcm-08-01249-t002:** Indications of emergency cesarean.

Indications	Seprafilm (*n*= 148)	Adept (*n* = 139)	*p*
Previous CS or uterine surgery	49	45	0.895
Breech	35	35	0.763
Prolonged labor	34	39	0.323
Fetal distress	34	26	0.375
Placenta previa	9	8	0.907
Macrosomia	4	0	0.051
Hypertensive disorder with complications	4	0	0.051
Elective	2	2	0.950
Extreme prematurity	1	0	0.332
Treatable fetal anomaly	0	1	0.301
Obstructive myoma	0	1	0.301

**Table 3 jcm-08-01249-t003:** Short-term postoperative outcomes.

Variable	Seprafilm (*n* = 148)	Adept (*n* = 139)	*p*
Puerperal fever ^†^	11 (7.4%)	8 (5.8%)	0.568
Postcesarean metritis	8 (5.4%)	4 (2.9%)	0.285
WBC ≥ 16,000 cells/μL	50 (33.8%)	50 (36.0%)	0.698
Band ≥ 10%	3 (2.0%)	0 (0.0%)	0.248
Incisional wound infection	9 (6.1%)	8 (5.8%)	0.789
Major complications ^#^	14 (9.5%)	11 (7.9%)	0.643
Length of hospital stay(days)	4.14 ± 0.51	4.13 ± 0.61	0.852
Flatus passage (h)	41.85 ± 11.483	36.95 ± 12.22	0.001 *
Flatus passage ≤48 h	90 (60.8%)	111 (79.9%)	<0.001 *
Flatus passage >48 h	58 (39.2%)	28 (20.1%)	
Use Dulcolax suppository	42 (28.4%)	24 (17.3%)	0.025 *
Early sip water	33 (22.3%)	40 (29.0%)	0.195
Use Menthol	2 (1.4%)	7 (5.0%)	0.095
Use Primperan IVD	2 (1.4%)	6 (4.3%)	0.161
Use Dimethicone	1 (0.7%)	2 (1.4%)	0.611

^†^ atelectasis, UTI, surgical-site infection (SSI), URI, mastitis, or breast abscess. ^#^ postpartum hemorrhage, chrioamnionitis, uterine rupture, abruptio placentae, sub-fascia hematoma, pulmonary edema, placentae accreta. * *p* < 0.05.

**Table 4 jcm-08-01249-t004:** Postcesarean metritis rate and laboratory data of contaminated and dirty wound, Seprafilm vs. Adept.

Variable	Seprafilm (*n* = 25)	Adept (*n* = 39)	*p*
Postcesarean metritis	8 (32.0%)	4 (10.3%)	0.048 *
WBC ≥ 16,000 cells/μL	12 (48.0%)	21 (58.3%)	0.648
Band ≥ 10%	1 (4.0%)	0 (0.0%)	0.391
Length of hospital stay	4.33 ± 1.08	4.28 ± 0.68	0.827
Incisional wound infection	3 (12.0%)	4 (10.3%)	1.000

* *p* < 0.05.

**Table 5 jcm-08-01249-t005:** Hazard ratio of postcesarean metritis.

Variable	Crude-OR (95% CI)	*p*	Adj-OR (95% CI) ^#^	*p*
Adept	1.00	0.037 *	1.00	0.015 *
Seprafilm	4.12 (1.09–15.61)		12.92 (1.64–101.53)	

^#^ After adjustment confounder of Age, GDM, DM SLE, Antibiotics, blood loss. * *p* < 0.05.

**Table 6 jcm-08-01249-t006:** Hazard ratio of flatus passage >48 h.

Variable	Crude-HR (95% CI)	*p*	Adj-HR (95% CI) ^#^	*p*
Adept	1	<0.001	1	0.001 *
Seprafilm	2.67 (1.52 to 4.44)		3.02 (1.70 to 5.36)	

^#^ Adjusted previous abdominal op, previous abdominal adhesion, sip water, Menthol, Primperan. IVD, Dimethicone, post OP fever. * *p* < 0.05.

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
