# Peer review of "Short-Term Postoperative Outcomes between 4% Icodextrin Solution and Hyaluronic Acid- Carboxymethyl Cellulose Membrane during Emergency Cesarean Section"

_jcm, 2019, doi:10.3390/jcm8081249_

Round 1
Reviewer 1 Report
The authors compare retrospectively Seprafilm and Adept as adhesion-reduction devices in emergency CS and found a better overall outcome by using Adept. The subject is interesting and novel, the article is well written and the methods are appropriate, even though the study design is retrospective in its nature. The discussion is proper.
Tables and Figure are well done.
I have a question: is it possible that the higher blood loss in the Seprafilm than in the Adept cases could influence the different performance of the two devices? Again: is there a reason for the higher blood loss in the Seprafilm group?
Reviewer 2 Report
Overall comments
In the abstract it is not clear what the study was performed. There is talk of a randomized study, but it is not clear what. Since the title focuses on two anti-adherents, Seprafilm and Adept, I think we can hypothesize that this is a comparison on these two devices.
The authors must review the type setting of the abstract, given that some words are combined (i.e. IN LINE 4: “outcomesof”).
The authors must be brief in transferring the results to the conclusions of the abstract. A reader who reads, in a nutshell, does not understand what the study is about.
In the introduction, authors also must review the type setting of the abstract, given that some words and square brackets are attached.
In the M&M line 56 there is a nonsense. If the study is retrospective, how do the authors recruit 287 patients and divide them into 2 groups?
If anything, the authors collected 287 patients retrospectively and analyzed all cases in which they used seprafilm and adept, excluding patients who did not meet the inclusion criteria. Then they divided the women with the devices into two groups. How then do the authors request consent to participate in the study if the study is retrospective? I also needed an opinion from the local IRB, but there is no mention in M&M.
There is something wrong with the study setting.
In the discussion, it would be appropriate to start with the major causes of fever after CS in an emergency, then discuss the topic of anti-adherence, the benefit of which, in my opinion, is marginal compared to multiple infectious problems. It should be anticipated the line 147 instead of line 138.
Personal comments
One thing that is not clear to me is why the authors use anti-adherents after a caesarean section, even if emergency, since the uselessness has already been proven.
